# AFMCC: Asynchronous Federated Multi-modal Constrained Clustering

## Abstract

Federated multi-modality clustering (FedMMC) aims to cluster distributed multi-modal data without compromising privacy. Existing approaches often rely on contrastive learning (CL), but suffer from representation degeneration, arbitrary modality missing, and computational imbalance. We propose Asynchronous Federated Multi-modal Constrained Clustering (AFMCC), which tackles these challenges through three key designs: (i) a Class-Correlation Matrix (CCM) regularization to prevent CL degeneration and enhance cluster separability, (ii) client-specific weighted aggregation to handle modality heterogeneity, and (iii) a weighted asynchronous aggregation strategy to mitigate computational imbalance and accelerate convergence. We further provide a theoretical analysis of AFMCC through a particle dynamics lens. Extensive experiments on diverse benchmarks demonstrate that AFMCC consistently outperforms state-of-the-art FedMMC methods in clustering accuracy and efficiency, while preserving privacy. We have released the source code and the dataset as supplementary material.

## 1 INTRODUCTION

Contrastive Learning (CL) is a self-supervised learning paradigm designed to learn high-quality data representations through similarity modeling. Its core idea is as follows: in the absence of explicit labels, sample pairs are divided into "positive pairs" and "negative pairs". The model is then trained to pull positive pairs closer while pushing negative pairs apart in the representation space. Over the past few years, CL has achieved remarkable progress across domains such as computer vision, natural language processing, and graph representation learning. Representative methods include SimCLR (Chen et al., 2020), MoCo (He et al., 2020), and BYOL (Grill et al., 2020). By combining effective data augmentation strategies with carefully designed contrastive loss functions, these approaches enable models to learn representations under unsupervised conditions that rival—or even surpass—those obtained with supervised training. In the context of federated learning, contrastive learning also demonstrates significant potential in the field of multimodal clustering: it not only enhances representation learning under multi-modality data fusion but also demonstrates robustness in scenarios with data heterogeneity and modality missingness (Chen et al., 2024).

Despite recent progress, applying CL to federated multimodal clustering remains non-trivial. Although CL has demonstrated potential in this domain, it still suffers from degeneration issues. As illustrated in Figure 1, CL often yields strong performance in the early stages of clustering but tends to deteriorate as training progresses. In practice, the inherent randomness of optimization makes it difficult to determine an appropriate stopping criterion, which substantially undermines clustering quality. Furthermore, existing approaches fall short of fully addressing the heterogeneity inherent in federated settings. For instance, FMCSC (Chen et al., 2024) partitions clients into single-modality (with only one modality) and multi-modality (with all modalities) groups, and employs mutual information to mitigate model discrepancies across clients. While this design achieves encouraging results, it is incapable of accommodating arbitrary missing-modality patterns (e.g., clients with two modalities coexisting alongside those with three). In addition, real-world federated systems often experience significant computational imbalance across clients. However, most current methods rely on synchronous training, which exacerbates inefficiencies and leads to underutilization of available resources.

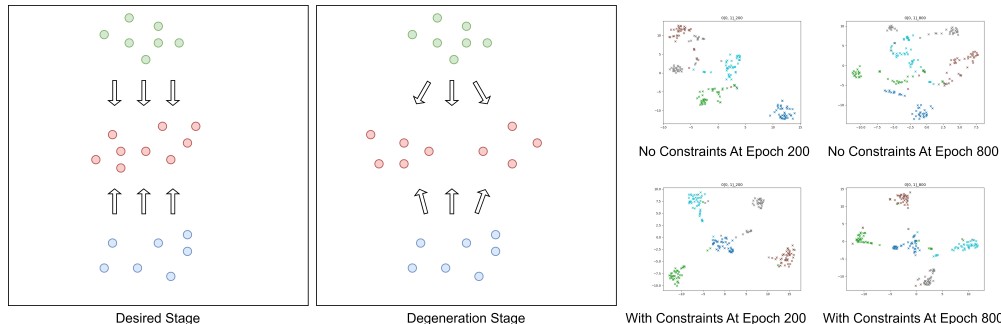

Figure 1: **Visualization of the Degeneration Process.** Left: two examples illustrating possible degeneration at later clustering stages. Right: The t-SNE visualization results (Glorot et al., 2011) illustrate the effect of our constraint analysis experiment on the BDGP dataset using FMCSC (see Section 4.2). The visualization demonstrates a comparison between training with and without the proposed constraints.

To address the above issues, we propose a series of solutions. To mitigate degeneration at later training stages, we introduce a constraint mechanism that maps the feature space into a pseudo-probability space and applies class-correlation matrices to regularize representations, effectively alleviating degeneration. To handle modality missingness, we design a deep neural network framework that leverages homogeneous models to process heterogeneous client data. For multimodal clients, we simulate missing modalities and encourage reconstruction, thereby enabling modality-deficient clients to extract representations enriched with full-modality information. To cope with computational imbalance, we propose an asynchronous federated learning method that allows training to proceed asynchronously, it can significantly reduce training time under computational imbalance.

By integrating these designs, we develop a unified framework termed **Asynchronous Multimodal Constrained Federated Clustering (AFMCC)**. As shown in Figure 2, clients interact with the server and update local parameters after each round of training (details in Section 3.5). This framework is capable of handling arbitrary patterns of missing modalities (one or multiple missing modalities, provided at least one modality is present), as elaborated in Section 3.2. Furthermore, we analyze the degeneration phenomenon in contrastive learning from a particle-dynamics perspective, as illustrated in Figure 1 and detailed in Section 3.3. We summarize our main contributions as follows:

- We provide a particle-dynamics perspective on degeneration in contrastive clustering. Building upon prior assumptions, we introduce a global constraint based on Class-Correlation Matrix (CCM) that effectively mitigates degeneration and enhances cluster separability and balance.

- We propose a novel federated learning framework for multimodal clustering under missing-modality conditions. The framework enables multimodal clients to assist clients with uni-modal or incomplete modalities, yielding superior global clustering performance. Additionally, we propose an asynchronous model aggregation strategy that significantly reduces training time under computational imbalance.

- We conduct both theoretical analyses and extensive experiments to demonstrate the effectiveness of AFMCC, showing consistent performance improvements across diverse scenarios.

## 2 RELATED WORKS

**Deep Multi-modal Clustering.** Traditional multi-modal clustering methods (e.g., subspace- or spectral-based approaches) emphasize explicit consistency or complementarity across modalities but are limited when handling high-dimensional, nonlinear representations. With the advent of deep representation learning, many works employ autoencoders, contrastive learning, or prototype-based methods to jointly learn cross-modal embeddings in low-dimensional spaces, achieving significant

clustering improvements (Xu et al., 2021; 2022; Wen et al., 2021). A common strategy is to fuse different modalities within a unified representation space. However, most approaches assume centralized data access or relatively complete modalities, limiting their applicability under privacy constraints or missing modalities.

**Partially Observed Modalities.** In real-world multimodal data, modality missingness is pervasive. Two mainstream strategies have emerged: (i) cross-modal generation or mapping, often via generative or adversarial models, to reconstruct missing modalities and enable downstream tasks in a shared embedding space; and (ii) modality-agnostic objectives that exploit available modalities through shared subspaces or collaborative training (Xu et al., 2019; Wang et al., 2023; Wen et al., 2024). While effective in centralized settings, these methods are difficult to transfer to federated environments, where privacy, communication cost, and inter-client distribution shifts pose additional challenges.

**Federated Clustering & Federated Multi-modal Learning.** Federated learning (FL) primarily addresses privacy-preserving training and communication efficiency. Recent extensions to clustering—so-called federated clustering—aim to learn global cluster structures or prototypes without exchanging raw data (Zhu et al., 2023; Che et al., 2023). Research on federated multimodal learning is relatively scarce; existing work typically focuses on secure parameter aggregation or cross-client representation alignment. However, few approaches jointly consider two practical challenges: (i) missing modalities and (ii) asynchronous updates under heterogeneous client resources. This gap limits current methods when tackling partially observed, heterogeneous, imbalanced computing resources and multimodal settings.

**Contrastive Learning Pitfalls & Regularization.** Contrastive learning has become a dominant paradigm for unsupervised representation learning (e.g., SimCLR, MoCo, SwAV), but it suffers from well-known degeneration issues, such as representation collapse, intra-/inter-class imbalance, and biased estimates under small batches or scarce negatives (Jing et al., 2021). To mitigate these problems, prior work incorporates prototype or cluster information, debiasing objectives, or entropy/balancing regularization (Caron et al., 2020). Nevertheless, these strategies are primarily designed for centralized or single-machine multimodal settings, and how to ensure both stability of contrastive objectives and discriminability of clusters under federated, missing-modality, and asynchronous conditions still under development.

**Asynchrony & System Heterogeneity in FL.** Clients in FL typically differ in computation, bandwidth, and participation frequency, making synchronous aggregation inefficient and resource-wasting. Asynchronous parameter servers, hierarchical aggregation, weighted delay compensation, and calibration-based updates have been proposed to address these challenges (Xie et al., 2019; Zheng et al., 2017). Yet most of these advances target supervised learning. When unsupervised contrastive or clustering objectives are involved, designing stable asynchronous aggregation, preserving representation consistency while avoiding cumulative noise, remains an open problem.

# 3 METHOD

In this section, we present our proposed framework AFMCC. As illustrated in Figure 2, each client consists of three main components: (i) a customized multi-modal autoencoder; (ii) a feature constraint module; (iii) a client-specific weighted aggregation mechanism. The central server stores and updates the parameters of all participating clients.

## 3.1 PROBLEM DEFINITION

We consider a heterogeneous federated learning setting with $U$ clients $\{C_1, C_2, \ldots, C_U\}$. Each client has access to at most $|\mathbb{V}_{all}|$ modalities, denoted by $\mathbb{V}_{all} = \{1, 2, \ldots, |\mathbb{V}_{all}|\}$. Client $u$ possesses a subset $\mathbb{V}_u \subseteq \mathbb{V}_{all}$, where $|\mathbb{V}_u| \leq |\mathbb{V}_{all}|$. Each client $u$ owns a private dataset of $N$ samples. A sample contains $|\mathbb{V}_u|$ modalities, represented as: $\boldsymbol{X}_u = \left\{ \{\boldsymbol{x}_{ui}^v : v \in \mathbb{V}_u\} \mid i = 1, \ldots, N \right\}$. where $\boldsymbol{x}_{ui}^v$ is the raw data of the $v$-th modality for sample $i$ at client $u$. The goal is to cluster these samples into $K$ balanced groups. The latent feature dimension is fixed to $d$. For simplicity, unless otherwise specified, we omit the client index $u$: e.g., $\boldsymbol{x}_i^v \equiv \boldsymbol{x}_{ui}^v$ and $\mathbb{V} \equiv \mathbb{V}_u$. We denote cosine similarity as $\text{sim}(\boldsymbol{a}, \boldsymbol{b}) = \frac{\boldsymbol{a} \cdot \boldsymbol{b}}{\|\boldsymbol{a}\| \|\boldsymbol{b}\|}$.

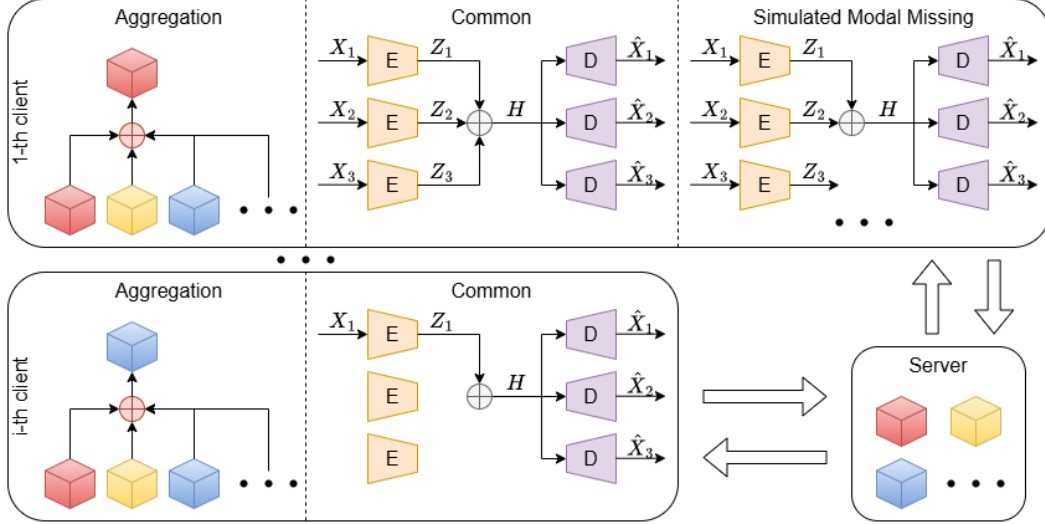

Figure 2: **The framework of AFMCC.** Assume at most three modalities per client: the first client possesses all three, while client $i$ has only one. After pretraining, (1) clients download parameters from others and aggregate them into new local models; (2) standard training proceeds locally; (3) multi-modal clients simulate missing-modality settings to aid incomplete-modality clients; (4) updated parameters are uploaded to the server. Steps (1)–(4) repeat until convergence.

## 3.2 MULTI-MODAL AUTOENCODER

Raw multimodal data often contains redundancy and noise. Self-supervised autoencoders, such as classical autoencoders (Hinton & Salakhutdinov, 2006) and variational autoencoders (Kingma & Welling, 2013), have shown strong capability in extracting compact latent features.

As shown in Figure 2, each client in AFMCC is equipped with a multi-modal autoencoder. The collection of client models is denoted as $\{f_1(\cdot; \boldsymbol{w}_1), f_2(\cdot; \boldsymbol{w}_2), \ldots, f_U(\cdot; \boldsymbol{w}_U)\}$, where $f_u(\cdot; \boldsymbol{w}_u)$ is the model of client $u$ with parameters $\boldsymbol{w}_u$. The $v$-th multi-modal autoencoder consists of modality-specific encoders and decoders: $E^v(\cdot; \boldsymbol{\theta}_1^v)$ and $D^v(\cdot; \boldsymbol{\theta}_2^v)$ for modality $v$, with learnable parameters $\boldsymbol{\theta}_1^v$ and $\boldsymbol{\theta}_2^v$. Encoders map raw data to latent representations: $\boldsymbol{z}_i^v = E^v(\boldsymbol{x}_i^v; \boldsymbol{\theta}_1^v)$. Given a modality subset $\mathbb{G} = \{g_1, g_2, \ldots, g_{|\mathbb{G}|}\} \subseteq \mathbb{V}$ (determined by the training scenario, e.g., simulating missing modalities; see Section 3.5), the fused feature is:

$$\boldsymbol{h}_i = \frac{1}{|\mathbb{G}|} \sum_{v \in \mathbb{G}} \boldsymbol{z}_i^v. \tag{1}$$

Decoders reconstruct modality-specific data from $\boldsymbol{h}_i$: $\hat{\boldsymbol{x}}_i^v = D^v(\boldsymbol{h}_i; \boldsymbol{\theta}_2^v)$. The autoencoder is trained by minimizing the reconstruction loss:

$$\mathcal{L}_r = \frac{1}{N} \sum_{v \in \mathbb{V}} \sum_{i=1}^{N} \|\boldsymbol{x}_i^v - \hat{\boldsymbol{x}}_i^v\|_2^2. \tag{2}$$

To enhance discriminability, we further adopt a contrastive objective that maximizes similarity between positive pairs (defined by matrix $A$) while pushing apart negatives:

$$\mathcal{L}_d = -\frac{1}{N} \sum_{v \in \mathbb{V}} \sum_{i=1}^{N} \log \frac{e^{\text{sim}(\boldsymbol{h}_i, \boldsymbol{z}_i^v)/\tau}}{\sum_{j \neq i} e^{\text{sim}(\boldsymbol{h}_i, \boldsymbol{z}_j^v)/\tau}}. \tag{3}$$

## 3.3 FEATURE CONSTRAINT METHOD

Contrastive learning (Chen et al., 2020; He et al., 2020; Grill et al., 2020) has attracted considerable attention in recent years and achieved remarkable success in clustering. We analyze its mechanism from two complementary perspectives.

From a **probabilistic perspective**, the spatial distance (or similarity) between the features of two samples can be directly linked to the probability that they form a positive pair: higher similarity indicates a greater likelihood of being a positive pair. Contrastive learning operationalizes this by treating multimodal features of the same sample as positive pairs and pulling them closer in the feature space, while treating features from different samples as negative pairs and pushing them apart. Through this process, the feature distance between samples with similar characteristics is reduced, thereby increasing the model's estimated probability of them being a positive pair, whereas dissimilar samples exhibit larger distances and correspondingly lower probabilities.

From a **particle dynamics perspective**, all features can be viewed as particles. Contrastive learning enforces attraction between particles of the same class and repulsion between those of different classes. In the early stages of clustering, particles are randomly distributed; forces are roughly isotropic, guiding each particle toward regions dominated by its class. However, once clusters form, repulsion among different clusters becomes anisotropic. Shearing forces may arise across clusters, splitting them apart and leading to Figure 1.

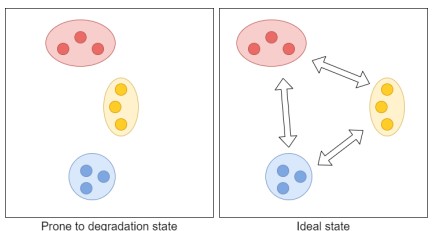 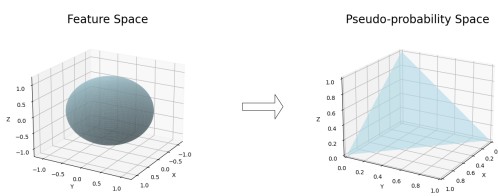

(a) Avoiding shear-induced splitting.      (b) Projecting features into a pseudo-probability space.

Figure 3: **A group of images.** Figure 3a shows a scenario prone to shear-induced splitting, which we aim to avoid, while the right image represents a more ideal situation. Figure 3b displays a 3-dimensional feature space and a 3-dimensional pseudo-probability space.

To address this issue, we propose a **feature constraint based on a class-correlation matrix**. Suppose the data are partitioned into $K$ classes of equal size $N/K$. As shown in Figure 3b, we project each feature into a pseudo-probability space using a learnable matrix $\boldsymbol{W} \in \mathbb{R}^{K \times d}$, and then define the **class-correlation matrix** as:

$$\boldsymbol{p}_i = \text{softmax}(\boldsymbol{W}\boldsymbol{h}_i), \qquad \boldsymbol{Q} = \frac{\boldsymbol{P}\boldsymbol{P}^\top}{N/K}, \tag{4}$$

where $\text{softmax}(\cdot)$ ensures that $\boldsymbol{p}_i \in \mathbb{R}^K$ is a valid probability distribution. Collecting all $\boldsymbol{p}_i$, we obtain a probability matrix $\boldsymbol{P} = [\boldsymbol{p}_1, \boldsymbol{p}_2, \ldots, \boldsymbol{p}_N] \in \mathbb{R}^{K \times N}$, with $P_{ij}$ denoting the probability that sample $i$ belongs to class $j$. Then, $\boldsymbol{Q} \in \mathbb{R}^{K \times K}$ captures correlations between class. Specifically, $Q_{ab} = \frac{K}{N} \sum_{i=1}^{N} P_{ai}P_{bi}$ measures the degree of correlation between $a$-th class and $b$-th class. Ideally, we want $\boldsymbol{Q}$ close to a target $\boldsymbol{Q}_{\text{tgt}}$. If $\boldsymbol{Q}_{\text{tgt}} = I_K$, each class would be perfectly separated with no cross-class correlation, yielding maximally discriminative clusters. However, this may over-constrain the representation, collapsing features into discrete points and discarding fine-grained contrastive structure. Instead, we relax the target to: $\boldsymbol{Q}_{\text{tgt}} = \lambda I_K + (1 - \lambda)\frac{1}{K}\boldsymbol{1}\boldsymbol{1}^\top$, Here, $\lambda$ serves as a flexibility coefficient: larger values of $\lambda$ enforce tighter clusters, whereas smaller values yield looser clusters. which preserves cluster separability while retaining sufficient flexibility for intra-class variability. Finally, we impose the following constraint loss:

$$\mathcal{L}_c = \|\boldsymbol{Q} - \boldsymbol{Q}_{\text{tgt}}\|_2^2, \tag{5}$$

which penalizes deviations of the empirical class-correlation matrix from the desired balanced structure. Intuitively, this constraint prevents shearing forces across clusters (see Figure 3a), thereby alleviating the degradation of contrastive learning in multi-modal clustering.

## 3.4 CLIENT-SPECIFIC WEIGHTED AGGREGATION

To bridge the gap between heterogeneous client models and enable clients with missing modalities to benefit from features of absent modalities—while also accommodating heterogeneous computational capacities—we propose an **asynchronous client-specific weighted aggregation scheme**.

After each round of training, a client uploads its current model parameters to the server and down-loads the parameters of other clients for local aggregation. The aggregated model then serves as the initialization for the next training round.

Specifically, after training, each client uploads its local model parameters to the server and down-loads other clients' models. For each downloaded model, the client extracts modality-specific features using all decoders and fuses them to obtain a feature set $\left\{ \left\{ \boldsymbol{Z}_u^v : v \in V_u \right\} \mid u = 1, \ldots, U \right\}$, where $\boldsymbol{Z}_u^v = [\boldsymbol{z}_{u1}^v, \boldsymbol{z}_{u2}^v, \ldots, \boldsymbol{z}_{uN}^v]$, where $\boldsymbol{z}_{ui}^v$ represents the feature extracted from the $i$-th local sample using client $u$'s decoder. We define the matrix as $\boldsymbol{H} = [\boldsymbol{h}_1, \boldsymbol{h}_2, \ldots, \boldsymbol{h}_N]$.

For each fused feature matrix, we compute a deviation score. The aggregation weight is then computed as:

$$\varepsilon_u = \sum_{v \in \mathbb{V}_u} \exp\left(\|\boldsymbol{Z}_u^v - \boldsymbol{H}\|_2\right), \qquad \boldsymbol{w}_{local} = \sum_{u=1}^{U} \frac{\varepsilon_u \boldsymbol{w}_u}{\sum_{i=1}^{U} \varepsilon_i}, \tag{6}$$

where $\varepsilon_u$ measures the alignment between features $\boldsymbol{Z}^v$ and the fused representation $\boldsymbol{H}$, and $\boldsymbol{w}_{local}$ is the aggregated model parameter assigned to the client.

### 3.5 ASYNCHRONOUS FEDERATED MULTI-MODAL CONSTRAINED CLUSTERING (AFMCC)

We now present the overall workflow of the proposed AFMCC framework, which is organized into three phases: **pretraining**, **parameter aggregation**, and **local training**.

**Pretraining phase.** The framework begins with a designated client conducting local training to obtain an initial model. Upon completion, the server disseminates the trained parameters to all other clients, providing a common initialization for subsequent training.

**Parameter aggregation phase.** Each client periodically downloads the latest parameters from the server and applies the weighting scheme defined in Eq. 6 to perform local aggregation. The aggregated parameters replace the existing local ones, thereby serving as the basis for further training. After each round of local updates, the client uploads its revised parameters to the server. Simultaneously, clients may also retrieve parameters from other participants and update their models through the same weighted aggregation scheme. This iterative process continues until convergence, which can be determined either by a fixed number of training rounds or by monitoring stability criteria (see Fig. 2).

**Local training phase.** During local training, clients with multiple modalities are required to learn not only from their full modality set $\mathbb{V}_i$, but also to simulate clients with missing modalities. For example, consider client $i$ with modality set $\mathbb{V}_i$ and client $j$ with $\mathbb{V}_j$, where $\mathbb{V}_j \subseteq \mathbb{V}_i$. In this case, client $i$ must ensure satisfactory performance on both $\mathbb{V}_i$ and $\mathbb{V}_j$. This is achieved by alternately setting the fusion set $G$ in Eq. 1 to $\mathbb{V}_i$ and $\mathbb{V}_j$ during training. Each client optimizes its model independently using local data, guided by the objective function:

$$\mathcal{L} = \mathcal{L}_r + \alpha \mathcal{L}_d + \beta \mathcal{L}_c, \tag{7}$$

where $\mathcal{L}_r$ denotes the reconstruction loss (preserving cross-modality consistency), $\mathcal{L}_d$ represents the discriminative loss (enhancing inter-class separability), and $\mathcal{L}_c$ corresponds to the contrastive constraint loss (facilitating the aggregation and separation of multi-modal features). Hyperparameters $\alpha$ and $\beta$ balance the relative contributions of these terms. Upon completion of local training, the updated parameters are transmitted to the server.

In summary, after the initial pretraining, each client alternates between parameter aggregation and local training. This iterative cycle continues until convergence, as defined by either a predetermined number of epochs or the stabilization of evaluation metrics (see Fig. 2). The pseudocode of the AFMCC procedure is provided in Algorithms 1 and 2.

## 4 EXPERIMENT

### 4.1 EXPERIMENTAL SETUP

We categorize clients into three types: **F-clients** (full-modality clients), which possess the union of all available modalities; **M-clients** (multi-modality clients), which own a subset of modalities

---

**Algorithm 1** AFMCC: Asynchronous Federated Multi-modal Constrained Clustering

---

1: ClientTrain($C_1$)                                                          ▷ Pretraining
2: **for** $i = 2, 3, \ldots, U$ **do**
3:     $\boldsymbol{w}_i \leftarrow \boldsymbol{w}_1$
4: **end for**
5: **for** $i = 1, 2, \ldots, U$ **do**
6:     **async** ClientTrain($C_i$)                   ▷ Asynchronous local training for all clients
7: **end for**

---

**Algorithm 2** ClientTrain: Local Training Procedure

---

1: **while** convergence not reached **do**
2:     Download model parameters $\{\boldsymbol{w}_1, \ldots, \boldsymbol{w}_U\}$ from the server
3:     Compute aggregated global parameters $\boldsymbol{w}_{\text{global}}$ via Eq. 6
4:     $\boldsymbol{w}_{\text{local}} \leftarrow \boldsymbol{w}_{\text{global}}$                  ▷ Initialize local model with global parameters
5:     **for** $\mathbb{G}$ in $\{\mathbb{A} \mid \mathbb{A} \subseteq \mathbb{V}, \ \mathbb{A} \neq \emptyset\}$ **do**
6:         Perform $\delta$ gradient update steps on Eq. 7 using $\mathbb{G}$ as the modality set
7:     **end for**
8:     Upload local model parameters $\boldsymbol{w}_{\text{local}}$ to the server
9: **end while**

---

with at least two present; and **S-clients** (single-modality clients), which contain only one modality. Experiments are conducted on four benchmark multi-modal datasets: **MNIST-USPS** (Peng et al., 2019), **BDGP** (Cai et al., 2012), **Multi-Fashion** (Xiao et al., 2017), and **NUSWIDE** (Chua et al., 2009).

**Baselines.** We compare AFMCC against ten state-of-the-art methods, including HCP-IMSC (Li et al., 2022), IMVC-CBG (Wang et al., 2022), DSIMVC (Tang & Liu, 2022), LSIMVC (Liu et al., 2022), ProImp (Li et al., 2023), JPLTD (Lv et al., 2023), CPSPAN (Jiang et al., 2024), FedDMVC (Chen et al., 2023), FCUIF (Ren et al., 2024), and FMCSC (Chen et al., 2024). Among them, FedDMVC, FCUIF, and FMCSC are federated multi-modal clustering methods, while the others are centralized incomplete multi-modal clustering approaches. For fair comparison, we concatenate client-partitioned data and feed it into centralized methods. In this setting, data from multi-modal clients are treated as complete, whereas data from single-modal clients are regarded as incomplete.

**Implementation details.** Each encoder follows the architecture Input–Fc512–Fc128–Fc20, and each decoder is symmetric to its encoder. Unless otherwise specified, experiments involving AFMCC are conducted on the BDGP dataset with an $F/S = 1 : 1$ client ratio. Default hyper-parameters are summarized in Table 1, where $d$ denotes the feature dimension, $\varphi$ the number of local gradient update steps, $\xi$ the number of communication rounds, and $\tau, \lambda, \alpha, \beta$ are fixed coefficients. Each experiment was independently repeated five times, and the average values were taken to ensure the robustness of the results.

## 4.2 EXPERIMENTS AND ANALYSIS

**Clustering results.** Following the experimental protocol of FMCSC, Table 2 reports the quantitative results under heterogeneous and mixed-modal scenarios. AFMCC consistently achieves superior clustering performance compared to existing methods.As the proportion of S-clients increases, all methods suffer a gradual decline in performance due to the lack of complete modality information. Nonetheless, AFMCC maintains strong clustering accuracy and robustness across varying client heterogeneity. These results highlight that AFMCC effectively balances privacy preservation and clustering quality, making it well-suited for real-world federated multi-modal settings.

**Complex clustering results.** Table 3a presents the experimental results of AFMCC on the Multi-Fashion dataset (Xiao et al., 2017) under more challenging heterogeneous scenarios. While alternative methods struggle to effectively address such complex settings, AFMCC consistently demonstrates superior clustering performance. Notably, AFMCC achieves competitive results even in the absence of F-clients (F/M/S = 0:1:1), underscoring both the robustness and broad applicability of the proposed framework.

Table 1: Default parameters. Unless otherwise specified, the following parameters are used in all experiments.

| Dataset | d | $\varphi$ | $\xi$ | $\tau$ | $\lambda$ | $\alpha$ | $\beta$ |
|---|---|---|---|---|---|---|---|
| MNIST-USPS | 20 | 5 | 50 | 0.5 | 0.3 | 0.10 | 0.01 |
| BDGP | 20 | 5 | 20 | 0.5 | 0.3 | 0.01 | 0.01 |
| Multi-Fashion | 20 | 5 | 200 | 0.5 | 0.3 | 1.00 | 0.01 |
| NUSWIDE | 20 | 5 | 300 | 0.5 | 0.1 | 1.00 | 0.01 |

Table 2: Clustering performance on heterogeneous multi-modal datasets. F/M/S indicates the ratio of full-modality, multi-modal, and single-modal clients. ACC, NMI, and ARI denote clustering accuracy, normalized mutual information, and adjusted Rand index. Best and second-best results are marked in bold and underline, respectively.

| | Method | F/S = 2:1 | | | F/S = 1:1 | | | F/S = 1:2 | | |
|---|---|---|---|---|---|---|---|---|---|---|
| | | ACC | NMI | ARI | ACC | NMI | ARI | ACC | NMI | ARI |
| MNIST-USPS | HCP-IMSC | 80.2±0.0 | 74.8±0.0 | 69.8±0.1 | 79.0±0.2 | 71.6±0.2 | 66.9±0.2 | 76.2±0.1 | 71.0±0.1 | 63.6±0.1 |
| | IMVC-CBG | 46.6±0.1 | 40.3±0.2 | 22.2±0.4 | 38.9±0.1 | 35.3±0.3 | 14.9±0.2 | 37.3±0.6 | 31.7±0.4 | 10.6±0.2 |
| | DSIMVC | 55.1±0.1 | 27.6±0.1 | 25.0±0.1 | 54.5±0.1 | 26.7±0.1 | 24.4±0.2 | 54.1±0.2 | 26.6±0.2 | 24.0±0.3 |
| | LSIMVC | 59.3±0.2 | 55.2±0.9 | 38.2±1.7 | 52.7±0.4 | 46.5±0.2 | 25.8±0.5 | 41.8±0.4 | 37.1±0.4 | 14.2±0.5 |
| | ProImp | 91.2±0.9 | 84.4±1.0 | 80.8±2.1 | 87.3±0.6 | 77.9±0.7 | 73.1±1.0 | 84.8±0.9 | 75.9±0.9 | 66.6±1.8 |
| | JPLTD | 40.7±0.1 | 22.6±0.1 | 17.3±0.0 | 40.0±0.2 | 19.8±0.1 | 15.2±0.1 | 32.3±0.1 | 13.7±0.1 | 10.1±0.1 |
| | CPSPAN | 79.5±2.5 | 77.3±2.0 | 70.7±2.2 | 76.5±2.7 | 73.7±2.1 | 66.6±3.0 | 74.6±3.6 | 74.5±3.3 | 64.5±3.5 |
| | FedDMVC | 81.1±0.9 | 81.9±0.9 | 73.7±1.4 | 69.9±1.5 | 72.5±2.9 | 58.9±2.6 | 63.1±0.6 | 62.6±0.5 | 48.4±0.3 |
| | FCUIF | 85.3±0.3 | 83.2±0.3 | 75.7±0.5 | 72.8±1.2 | 70.3±1.5 | 64.4±1.8 | 67.2±0.8 | 64.4±0.9 | 53.5±0.8 |
| | FMCSC | 95.1±0.8 | 87.8±1.2 | 88.8±1.5 | 92.9±1.2 | 84.2±2.2 | 85.0±2.5 | 90.1±1.2 | 79.4±2.6 | 79.5±3.2 |
| | AFMCC | **98.0±0.0** | **94.5±0.1** | **95.5±0.1** | **96.5±0.0** | **91.2±0.1** | **92.3±0.1** | **94.3±0.1** | **86.9±0.2** | **87.8±0.3** |
| BDGP | HCP-IMSC | 93.1±0.0 | 81.9±0.0 | 83.6±0.0 | 89.8±0.0 | 73.4±0.1 | 76.4±0.0 | 89.5±0.0 | 72.5±0.1 | 76.7±0.0 |
| | IMVC-CBG | 37.9±0.4 | 21.0±1.0 | 10.4±0.1 | 37.2±0.1 | 21.0±0.0 | 7.8±0.0 | 36.9±0.1 | 20.4±0.1 | 6.4±0.0 |
| | DSIMVC | 92.5±0.4 | 81.7±0.8 | 84.9±0.9 | 89.5±2.0 | 76.5±2.0 | 77.8±2.2 | 86.1±3.3 | 70.0±3.9 | 76.6±3.4 |
| | LSIMVC | 44.1±0.5 | 23.7±0.4 | 5.7±0.2 | 39.2±1.3 | 19.7±0.5 | 4.8±0.2 | 35.3±1.6 | 14.9±1.3 | 2.8±0.4 |
| | ProImp | 91.6±0.3 | 82.4±3.8 | 80.0±0.9 | 90.4±1.5 | 76.2±0.5 | 79.3±1.6 | 75.6±0.5 | 52.3±2.0 | 44.6±1.8 |
| | JPLTD | 56.5±0.2 | 41.3±0.1 | 31.7±0.0 | 49.4±0.1 | 33.3±0.0 | 18.5±0.0 | 51.0±0.2 | 34.1±0.1 | 21.5±0.0 |
| | CPSPAN | 78.7±0.6 | 58.3±1.3 | 58.6±1.6 | 57.3±1.3 | 50.3±2.3 | 39.4±3.7 | 52.4±1.5 | 34.7±1.4 | 27.1±2.1 |
| | FedDMVC | 92.0±0.1 | 80.2±0.2 | 84.7±0.1 | 91.5±0.5 | 77.1±0.4 | 80.3±0.7 | 82.2±0.2 | 63.4±0.3 | 61.9±0.3 |
| | FCUIF | 93.8±0.1 | 82.2±0.1 | 85.1±0.1 | 90.3±0.2 | 75.2±0.3 | 78.4±0.3 | 85.7±0.2 | 67.5±0.3 | 63.2±0.2 |
| | FMCSC | 94.5±0.8 | 83.9±1.2 | 86.8±1.5 | 91.9±1.2 | 77.3±2.2 | 81.0±2.5 | 90.0±1.2 | 73.3±2.6 | 76.8±3.2 |
| | AFMCC | **95.7±0.4** | **87.2±2.4** | **89.7±2.0** | **93.9±0.2** | **82.3±1.0** | **85.4±1.0** | **90.7±0.6** | **75.3±1.9** | **78.3±2.3** |
| Multi-Fashion | HCP-IMSC | 70.6±0.1 | 67.4±0.1 | 57.7±0.1 | 67.1±0.1 | 64.7±0.1 | 53.1±0.1 | 59.9±0.7 | 56.4±0.9 | 41.8±1.1 |
| | IMVC-CBG | 46.3±0.0 | 49.4±0.0 | 26.3±0.0 | 43.2±0.1 | 42.7±0.1 | 19.2±0.1 | 38.9±0.2 | 39.4±0.3 | 13.5±0.4 |
| | DSIMVC | 82.7±1.3 | 83.6±1.1 | 74.5±1.1 | 77.7±1.4 | 76.7±0.8 | 66.8±0.8 | 76.7±1.7 | 75.8±1.5 | 66.4±1.4 |
| | LSIMVC | 51.1±0.5 | 49.9±0.1 | 31.5±0.5 | 50.2±0.6 | 52.2±0.1 | 35.2±0.1 | 49.9±0.2 | 48.6±0.0 | 28.2±0.0 |
| | ProImp | 69.1±0.4 | 66.3±0.3 | 55.2±0.8 | 69.0±0.1 | 64.6±0.2 | 52.5±0.2 | 53.9±2.4 | 50.7±1.5 | 27.8±1.6 |
| | JPLTD | 44.6±0.0 | 43.4±0.0 | 36.9±0.1 | 37.2±0.1 | 36.6±0.1 | 29.5±0.1 | 25.8±0.1 | 25.1±0.1 | 16.5±0.1 |
| | CPSPAN | 64.1±1.2 | 71.4±1.3 | 55.8±1.5 | 61.6±2.0 | 69.7±1.3 | 54.4±1.8 | 59.3±2.0 | 68.0±1.8 | 53.3±1.9 |
| | FedDMVC | 67.7±0.3 | 74.6±0.8 | 58.0±0.6 | 66.6±0.4 | 65.3±0.7 | 54.3±0.7 | 57.6±0.7 | 58.5±0.8 | 43.2±0.7 |
| | FCUIF | 70.7±0.5 | 79.4±0.9 | 63.1±0.4 | 68.4±0.6 | 71.5±0.4 | 59.2±0.5 | 62.5±0.6 | 61.3±0.5 | 45.6±0.5 |
| | FMCSC | 92.4±0.1 | 85.8±0.2 | 84.7±0.3 | 90.4±0.6 | 82.8±0.7 | 80.9±1.0 | 87.5±0.6 | 79.1±0.1 | 76.3±1.0 |
| | AFMCC | **93.4±0.0** | **87.7±0.1** | **86.5±0.1** | **91.8±0.0** | **84.5±0.0** | **83.5±0.1** | **87.8±0.1** | **79.3±0.1** | **76.5±0.2** |
| NUSWIDE | HCP-IMSC | 36.1±0.0 | 8.5±0.0 | 6.7±0.0 | 35.3±0.1 | 8.2±0.0 | 6.3±0.0 | 31.3±0.0 | 6.0±0.1 | 4.7±0.1 |
| | IMVC-CBG | 30.8±0.1 | 4.8±0.0 | 3.1±0.0 | 30.4±0.0 | 4.6±0.0 | 2.5±0.0 | 29.3±0.0 | 4.0±0.0 | 1.9±0.1 |
| | DSIMVC | 51.1±1.3 | 25.3±0.8 | 23.4±0.8 | 50.6±0.8 | 22.2±0.7 | 20.4±0.6 | 46.7±0.5 | 18.3±0.6 | 16.0±0.4 |
| | LSIMVC | 37.2±0.2 | 10.8±0.1 | 6.8±0.1 | 36.4±0.3 | 11.8±0.1 | 7.0±0.4 | 33.9±0.4 | 9.2±0.3 | 5.8±0.2 |
| | ProImp | 38.4±0.1 | 11.1±0.0 | 8.3±0.1 | 37.1±0.4 | 10.5±0.1 | 7.6±0.2 | 34.3±0.7 | 8.0±0.0 | 6.1±0.0 |
| | JPLTD | 53.0±0.2 | 25.9±0.2 | 23.7±0.1 | 51.5±0.5 | 22.5±1.0 | 21.5±0.8 | 50.0±0.2 | 19.4±0.1 | 14.1±0.1 |
| | CPSPAN | 33.7±0.2 | 9.0±0.9 | 6.2±0.1 | 33.3±0.3 | 6.6±0.9 | 4.4±0.1 | 29.4±0.6 | 5.4±1.1 | 3.2±0.4 |
| | FedDMVC | 41.7±0.2 | 14.4±0.1 | 12.3±0.1 | 37.5±0.7 | 9.8±0.9 | 7.8±0.5 | 32.6±0.2 | 5.8±0.2 | 4.3±0.2 |
| | FCUIF | 45.2±0.3 | 15.0±0.3 | 14.1±0.2 | 40.2±0.5 | 10.0±0.6 | 9.2±0.5 | 38.2±0.4 | 9.6±0.3 | 8.2±0.3 |
| | FMCSC | 56.1±0.2 | 26.3±0.5 | 23.9±0.4 | 52.7±0.2 | 23.0±0.3 | 21.8±0.4 | 50.8±0.9 | 20.1±0.7 | 18.8±0.8 |
| | AFMCC | **58.5±0.5** | **28.7±0.2** | **27.6±0.9** | **54.5±0.1** | **23.6±0.1** | **22.3±0.1** | **51.7±0.9** | **20.3±1.0** | **18.9±0.8** |

**Ablation studies.** Component A denotes the contrastive learning loss, Component B denotes the constraint loss, and Component C denotes the client-specific weighted aggregation. Table 3b investigates the influence of these three components on AFMCC. When Component A is removed, the clustering accuracy drops drastically, indicating that contrastive learning is indispensable and serves as one of the core elements of AFMCC. Removing Component B results in only a marginal accuracy decline, as the model does not yet suffer from severe degeneration, which is consistent with our expectation. Eliminating Component C, i.e., replacing the client-specific weighted aggregation

Table 3: Experimental results.

| F/M/S | ACC | NMI | ARI |
|---|---|---|---|
| **1:1:1** | 91.8 | 85.1 | 83.6 |
| **0:1:1** | 87.1 | 89.5 | 75.6 |
| **1:1:0** | 96.2 | 92.4 | 92.0 |

(a) Complex clustering.

| A | B | C | ACC | NMI | ARI |
|---|---|---|---|---|---|
|  | ✓ | ✓ | 55.5 | 25.5 | 29.9 |
| ✓ |  | ✓ | 93.0 | 83.5 | 80.8 |
| ✓ | ✓ |  | 83.4 | 64.9 | 64.8 |

(b) Ablation.

| Step | UC | HC |
|---|---|---|
| **200** | 90.58 | 91.06 |
| **800** | 81.65 | 90.01 |

(c) Constraint.

| E/O | ACC | NMI | ARI |
|---|---|---|---|
| **2:1** | 93.3 | 84.1 | 81.1 |
| **3:1** | 91.9 | 81.2 | 78.8 |

(d) Asynchronous.

with a uniform aggregation (equal weights for all clients), leads to a notable accuracy decrease, demonstrating the superiority of our aggregation strategy.

**Parameter analysis.** The experimental results are presented in Figure 4. illustrates the effects of $\alpha$, $\beta$, and $\lambda$ on AFMCC. We observe that when $\alpha$ and $\beta$ are set to relatively small values, the clustering performance of the proposed AFMCC method degrades, which may be attributed to an overemphasis on data reconstruction that hinders the extraction of common features. Conversely, when $\alpha$ and $\beta$ are set to large values, clustering performance also declines, likely due to an excessive focus on view consistency that makes it difficult to disentangle the intrinsic feature space. In contrast, the method exhibits insensitivity to variations in $\lambda$: within the range of 0.2 to 1, the clustering accuracy fluctuates by no more than 0.8.

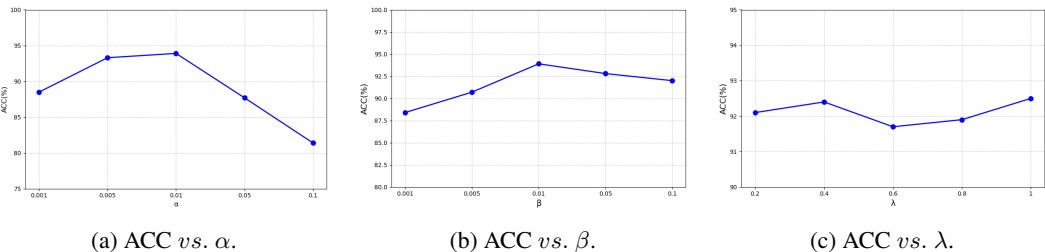

(a) ACC $vs.$ $\alpha$.  (b) ACC $vs.$ $\beta$.  (c) ACC $vs.$ $\lambda$.

Figure 4: Parameter analysis.

**Asynchronous analysis.** Table 3d presents the experimental results under computationally imbalanced settings. Here, $E/O$ denotes the ratio of computing power between even-indexed clients and odd-indexed clients, and the termination condition is defined as each odd-indexed client completing 10 training rounds. The results demonstrate that AFMCC effectively adapts to federated learning scenarios with heterogeneous computational capabilities and can fully exploit the available client resources.

**Constraint analysis.** To assess the effectiveness of our constraint mechanism, we extend FMCSC by incorporating the constraint loss into its objective, i.e., employing $\mathcal{L}_{pre} + 0.01\mathcal{L}_c(\lambda = 0.3)$, where $\mathcal{L}_{pre}$ denotes the original loss function. Experiments are conducted on the BDGP dataset with training steps set to 200 and 800, while other parameters follow the default settings in FMCSC (Chen et al., 2024). The results are summarized in Table 3c, where "UC" indicates the absence of the constraint loss and "HC" denotes the presence of it. In addition, Figure 1 illustrates the t-SNE visualization (Glorot et al., 2011) of the learned embeddings. These findings verify that our proposed constraint strategy effectively mitigates the degeneration problem commonly observed in contrastive learning within multi-modal federated clustering.

## 5 CONCLUSION

We introduced AFMCC, an asynchronous multi-modal federated clustering framework addressing modality missingness, heterogeneity, and straggler effects. By combining a CCM regularizer with client-weighted aggregation, AFMCC achieves stability and avoids degeneration, supported by our theoretical analysis. Experiments on four benchmarks show consistent gains, robustness to partial modalities, and faster convergence under asynchrony. Limitations include hyperparameter sensitivity and fixed cluster number; future work will explore adaptive schedules, non-parametric clustering, and stronger privacy guarantees.

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

# CONTENTS OF APPENDIX

# A  ANALYZING CONTRASTIVE LEARNING FROM A PARTICLE DYNAMICS PERSPECTIVE

## A.1  SETUP AND ASSUMPTIONS

We adopt the following assumptions:

- **Normalized embeddings.** $\|\boldsymbol{h}_i\| = \|\boldsymbol{z}_j^v\| = 1$, and similarity is measured by cosine similarity: $\mathrm{sim}(\boldsymbol{a}, \boldsymbol{b}) = \boldsymbol{a}^\top \boldsymbol{b}$.

- **Temperature parameter.** The temperature $\tau > 0$ is fixed, and the learning rate is sufficiently small so that optimization can be approximated by a gradient flow.

- **Probability mapping and class-related matrix.**

$$\boldsymbol{P} = \left[\boldsymbol{p}_i\right]_{i=1}^N \in \mathbb{R}^{K \times N}, \qquad \boldsymbol{Q} = \frac{\boldsymbol{P}\boldsymbol{P}^\top}{N/K} \in \mathbb{R}^{K \times K}.$$

The target matrix is set to

$$\boldsymbol{Q}_{\mathrm{tgt}} = \lambda_{\mathrm{tgt}}\boldsymbol{I}_K + (1 - \lambda_{\mathrm{tgt}})\tfrac{1}{K}\mathbf{1}\mathbf{1}^\top.$$

## A.2 FROM CONTRASTIVE LOSS TO POTENTIAL AND FORCES

When the sample size is sufficiently large, we can approximately view the contrastive loss as

$$\mathcal{L}_{\mathrm{d}} \approx \mathcal{L}_{\mathrm{CL}} = -\frac{1}{N} \sum_{v \in \mathbb{V}} \sum_{i=1}^{N} \log \frac{\exp(\boldsymbol{h}_i^\top \boldsymbol{z}_i^v / \tau)}{\sum_{j=1}^{N} \exp(\boldsymbol{h}_i^\top \boldsymbol{z}_j^v / \tau)}.$$

Define the similarity score as

$$s_{ij}^v = \frac{\boldsymbol{h}_i^\top \boldsymbol{z}_j^v}{\tau},$$

so that each sub-loss can be rewritten as

$$\ell_i^v = -\log \frac{e^{s_{ii}^v}}{\sum_j e^{s_{ij}^v}} = -s_{ii}^v + \log \sum_j e^{s_{ij}^v}.$$

Taking derivatives with respect to $s_{ij}^v$, we obtain

$$\frac{\partial \ell_i^v}{\partial s_{ij}^v} = \pi_{ij}^v - 1\{j = i\},$$

where

$$\pi_{ij}^v = \frac{e^{s_{ij}^v}}{\sum_k e^{s_{ik}^v}}, \qquad \sum_j \pi_{ij}^v = 1.$$

Intuitively, $\pi_{ij}^v$ represents the probability that anchor $i$ regards sample $j$ as a match in modality $v$. Thus, the gradient equals the softmax probability minus the one-hot ground truth label—precisely the standard form of cross-entropy.

Since $s_{ij}^v = \frac{1}{\tau} \boldsymbol{h}_i^\top \boldsymbol{z}_j^v$, it follows that

$$\frac{\partial s_{ij}^v}{\partial \boldsymbol{h}_i} = \frac{1}{\tau} \boldsymbol{z}_j^v, \qquad \frac{\partial s_{ij}^v}{\partial \boldsymbol{z}_j^v} = \frac{1}{\tau} \boldsymbol{h}_i.$$

Hence,

$$\nabla_{\boldsymbol{h}_i} \ell_i^v = \sum_j \frac{\partial \ell_i^v}{\partial s_{ij}^v} \frac{\partial s_{ij}^v}{\partial \boldsymbol{h}_i} = \frac{1}{\tau} \Big( \sum_j \pi_{ij}^v \boldsymbol{z}_j^v - \boldsymbol{z}_i^v \Big),$$

$$\nabla_{\boldsymbol{z}_j^v} \ell_i^v = \frac{\partial \ell_i^v}{\partial s_{ij}^v} \frac{\partial s_{ij}^v}{\partial \boldsymbol{z}_j^v} = \frac{1}{\tau} \big( \pi_{ij}^v - 1\{j = i\} \big) \boldsymbol{h}_i.$$

Summing over all modalities $v$ and samples $i$, and dividing by $N$, the gradient of the total loss $\mathcal{L}_{\mathrm{CL}}$ is

$$\nabla_{\boldsymbol{h}_i} \mathcal{L}_{\mathrm{CL}} = \frac{1}{N\tau} \sum_v \Big( \sum_j \pi_{ij}^v \boldsymbol{z}_j^v - \boldsymbol{z}_i^v \Big), \qquad \nabla_{\boldsymbol{z}_j^v} \mathcal{L}_{\mathrm{CL}} = \frac{1}{N\tau} \sum_i \big( \pi_{ij}^v - 1\{i = j\} \big) \boldsymbol{h}_i.$$

Defining the "forces" as the negative gradients, we have

$$\mathbf{F}_i^{(\mathrm{CL}, \boldsymbol{h})} = \frac{1}{N\tau} \sum_v \boldsymbol{z}_i^v - \frac{1}{N\tau} \sum_v \sum_j \pi_{ij}^v \boldsymbol{z}_j^v,$$

$$\mathbf{F}_j^{(\mathrm{CL}, \boldsymbol{z}, v)} = \frac{1}{N\tau} \boldsymbol{h}_j - \frac{1}{N\tau} \sum_i \pi_{ij}^v \boldsymbol{h}_i.$$

Thus, the contrastive loss naturally decomposes into two components: an attractive force from positive pairs and a repulsive force from negative pairs.

### A.3 ELASTIC FORCES FROM CLASS-MATRIX CONSTRAINTS

**Gradient derivation of the class-matrix constraint.** Consider the constraint loss

$$\mathcal{L}_{\mathrm{C}} = \big\| \boldsymbol{Q} - \boldsymbol{Q}_{\mathrm{tgt}} \big\|_F^2, \tag{8}$$

with

$$\boldsymbol{Q} = \frac{K}{N} \sum_{i=1}^N \boldsymbol{p}_i (\boldsymbol{p}_i)^\top, \qquad \boldsymbol{p}_i = \mathrm{softmax}(\boldsymbol{u}_i), \quad \boldsymbol{u}_i = \boldsymbol{W}\boldsymbol{h}_i. \tag{9}$$

Let $\boldsymbol{A} := \boldsymbol{Q} - \boldsymbol{Q}_{\mathrm{tgt}}$. Using $\partial \|\boldsymbol{A}\|_F^2 / \partial \boldsymbol{A} = 2\boldsymbol{A}$, we obtain

$$\frac{\partial \mathcal{L}_{\mathrm{C}}}{\partial \boldsymbol{Q}} = 2\boldsymbol{A}. \tag{10}$$

A perturbation of a single probability vector $\boldsymbol{p}_i \mapsto \boldsymbol{p}_i + d\boldsymbol{p}_i$ induces

$$d\boldsymbol{Q} = \frac{K}{N} \big( d\boldsymbol{p}_i (\boldsymbol{p}_i)^\top + \boldsymbol{p}_i (d\boldsymbol{p}_i)^\top \big). \tag{11}$$

Thus the differential of the loss contributed is

$$d\mathcal{L}_{\mathrm{C}} = \big\langle \frac{\partial \mathcal{L}_{\mathrm{C}}}{\partial \boldsymbol{Q}}, d\boldsymbol{Q} \big\rangle_F = \frac{2K}{N} \Big( \mathrm{Tr}\big( (\boldsymbol{A})^\top d\boldsymbol{p}_i (\boldsymbol{p}_i)^\top \big) + \mathrm{Tr}\big( (\boldsymbol{A})^\top \boldsymbol{p}_i (d\boldsymbol{p}_i)^\top \big) \Big)$$

$$= \frac{2K}{N} \Big( (d\boldsymbol{p}_i)^\top \boldsymbol{A}\boldsymbol{p}_i + (d\boldsymbol{p}_i)^\top (\boldsymbol{A})^\top \boldsymbol{p}_i \Big). \tag{12}$$

Noting that $\boldsymbol{A}$ is symmetric (since $\boldsymbol{Q}$ and $\boldsymbol{Q}_{\mathrm{tgt}}$ are symmetric), the two terms are equal and we get

$$d\mathcal{L}_{\mathrm{C}} = \frac{4K}{N} (d\boldsymbol{p}_i)^\top \boldsymbol{A}\boldsymbol{p}_i. \tag{13}$$

Hence the gradient with respect to the probability vector is

$$\frac{\partial \mathcal{L}_{\mathrm{C}}}{\partial \boldsymbol{p}_i} = \frac{4K}{N} \boldsymbol{A}\boldsymbol{p}_i \tag{14}$$

To obtain the gradient with respect to the embedding $\boldsymbol{h}_i$, use the chain rule $\nabla_{\boldsymbol{h}_i} \mathcal{L}_{\mathrm{C}} = \big( \frac{\partial \boldsymbol{p}_i}{\partial \boldsymbol{h}_i} \big)^\top \frac{\partial \mathcal{L}_{\mathrm{C}}}{\partial \boldsymbol{p}_i}$. For the softmax one has the Jacobian

$$J(\boldsymbol{p}) := \frac{\partial \mathrm{softmax}(\boldsymbol{u})}{\partial \boldsymbol{u}} = \mathrm{diag}(\boldsymbol{p}) - \boldsymbol{p}\boldsymbol{p}^\top, \tag{15}$$

and $\partial \boldsymbol{u}/\partial \boldsymbol{h} = \boldsymbol{W}$. Therefore

$$\nabla_{\boldsymbol{h}_i} \mathcal{L}_{\mathrm{C}} = \frac{4K}{N} (\boldsymbol{W})^\top J(\boldsymbol{p}_i) (\boldsymbol{Q} - \boldsymbol{Q}_{\mathrm{tgt}}) \boldsymbol{p}_i \tag{16}$$

Finally, the corresponding "elastic force" (as defined in the text) is

$$\mathbf{F}_i^{(\mathrm{C}, \boldsymbol{h})} = -\nabla_{\boldsymbol{h}_i} \mathcal{L}_{\mathrm{C}} = -\frac{4K}{N} (\boldsymbol{W})^\top J(\boldsymbol{p}_i) (\boldsymbol{Q} - \boldsymbol{Q}_{\mathrm{tgt}}) \boldsymbol{p}_i \tag{17}$$

### A.4 TOTAL DYNAMICS AND FORCE DECOMPOSITION

Under normalization, the actual dynamics are given by the tangential projection:

$$\dot{\boldsymbol{x}} = \Pi_{\boldsymbol{x}} \mathbf{F}(\boldsymbol{x}), \quad \Pi_{\boldsymbol{x}} = \boldsymbol{I} - \boldsymbol{x}\boldsymbol{x}^\top.$$

Let $\mathcal{L}_{\mathrm{tot}} = \mathcal{L}_{\mathrm{CL}} + \gamma \mathcal{L}_{\mathrm{C}}$. The resulting dynamics can be expressed as

$$\dot{\boldsymbol{h}}_i = \Pi_{\boldsymbol{h}_i} \Big( \mathbf{F}_i^{(\mathrm{CL}, \boldsymbol{h})} + \gamma \, \mathbf{F}_i^{(\mathrm{C}, \boldsymbol{h})} \Big), \quad \dot{\boldsymbol{z}}_j^v = \Pi_{\boldsymbol{z}_j^v} \Big( \mathbf{F}_j^{(\mathrm{CL}, \boldsymbol{z}, v)} + \gamma \, \mathbf{F}_j^{(\mathrm{C}, \boldsymbol{z}, v)} \Big).$$

Intuitively, we arrive at the **three-way decomposition of forces**:

$$\mathbf{F} = \underbrace{\mathbf{F}_{\mathrm{attr}}}_{\text{positive attraction}} + \underbrace{\mathbf{F}_{\mathrm{rep}}}_{\text{negative repulsion}} + \underbrace{\mathbf{F}_{\mathrm{elas}}}_{\text{elastic restoration from class constraints}}.$$

**Interpretation of the force decomposition.** The total force $\mathbf{F}$ acting on each embedding can be understood as the superposition of three distinct components, each of which arises from a different term in the loss function and plays a complementary role in shaping the geometry of the learned representation space.

- **Positive attraction** ($\mathbf{F}_{\text{attr}}$). This term originates from the contribution of positive pairs in the contrastive loss. It acts as a pulling force that encourages embeddings of matched pairs (e.g., across different modalities or augmentations of the same sample) to align closely in the latent space. Mathematically, it corresponds to the gradient component $-\boldsymbol{z}_i^v/\tau$, which drives the anchor $\boldsymbol{h}_i$ toward its positive counterpart $\boldsymbol{z}_i^v$. Without this term, the model would fail to enforce semantic consistency across views.

- **Negative repulsion** ($\mathbf{F}_{\text{rep}}$). In contrast, this force is induced by the presence of negative pairs. It prevents embeddings from collapsing by pushing different samples away from each other. Concretely, it corresponds to the expectation over the softmax weights $\pi_{ij}^v$, which distribute repulsion strength according to similarity scores. This soft competition ensures that negatives that are more easily confused with the anchor exert stronger repulsive forces, thereby improving discriminability in the embedding space.

- **Elastic restoration** ($\mathbf{F}_{\text{elas}}$). Beyond pairwise attraction and repulsion, the class-matrix constraint introduces a global regularization force. As derived in Eq. 16, this term acts like an "elastic spring" that pulls the empirical class co-occurrence matrix $\boldsymbol{Q}$ toward the target structure $\boldsymbol{Q}_{\text{tgt}}$. Geometrically, this enforces a balanced allocation of embeddings across classes and prevents degenerate solutions such as class collapse. The force is mediated by the Jacobian of the softmax, meaning that restoration strength adapts to the confidence distribution of each sample's predicted class.

Overall, this three-way decomposition highlights the particle-dynamics analogy: embeddings behave like particles subject to competing forces of attraction, repulsion, and elastic restoration. The interplay of these forces stabilizes representation learning, ensuring both local consistency (via $\mathbf{F}_{\text{attr}}$ and $\mathbf{F}_{\text{rep}}$) and global structure alignment (via $\mathbf{F}_{\text{elas}}$).

## A.5 ANTI-COLLAPSE ENERGY GAP THEOREM

**Theorem 1** (Anti-Collapse Energy Gap). *Let $\lambda_{\text{tgt}} \in (0, 1]$. If all samples collapse in the sense that $\boldsymbol{p}_1 = \cdots = \boldsymbol{p}_N = \boldsymbol{p}$, then*

$$\boldsymbol{Q} = K\,\boldsymbol{p}\boldsymbol{p}^\top, \qquad \mathcal{L}_{\text{C}} = \|K\,\boldsymbol{p}\boldsymbol{p}^\top - \boldsymbol{Q}_{\text{tgt}}\|_F^2.$$

*In particular, unless $\boldsymbol{p}$ matches the target distribution exactly (so that $\boldsymbol{Q} = \boldsymbol{Q}_{\text{tgt}}$), one has $\mathcal{L}_{\text{C}} > 0$. On the other hand, there exist non-degenerate solutions such that $\mathcal{L}_{\text{C}} = 0$. Therefore, the fully collapsed solution cannot be globally optimal.*

## A.6 LOCAL LINEARIZATION

In the neighborhood of an anchor point $\boldsymbol{h}_i$, the derivative of the softmax weight $\pi_{ij}^v$ with respect to $\boldsymbol{h}_i$ is

$$\frac{\partial \pi_{ij}^v}{\partial \boldsymbol{h}_i} = \frac{1}{\tau} J(\pi_i^v)(\boldsymbol{z}_j^v), \qquad J(\pi) = \text{diag}(\pi) - \pi\pi^\top.$$

Substituting this yields the local Jacobian approximation:

$$\mathrm{D}\dot{\boldsymbol{h}}_i \ \approx \ -\frac{1}{N\tau^2} \sum_v \left( \sum_j \pi_{ij}^v \boldsymbol{z}_j^v \boldsymbol{z}_j^{v\top} - \left(\sum_j \pi_{ij}^v \boldsymbol{z}_j^v\right)\left(\sum_j \pi_{ij}^v \boldsymbol{z}_j^v\right)^\top \right) - \gamma\,\mathcal{K}_i, \tag{18}$$

where $\mathcal{K}_i$ denotes the Hessian of the class-constraint term with respect to $\boldsymbol{h}_i$. The first term includes an additional rank-one correction $-\boldsymbol{m}\boldsymbol{m}^\top$ with $\boldsymbol{m} = \sum_j \pi_{ij}^v \boldsymbol{z}_j^v$.

## A.7 STABILITY CONDITION FOR SHEAR-SPLITTING RESISTANCE

From Eq. 18, stability is determined by the maximum eigenvalue restricted to the tangent subspace. We obtain the following result:

**Proposition 1** (Shear-Splitting Resistance under Approximation). *If for all $i$ there exists a constant $c_{\text{pos}} > 0$ such that $\lambda_{\min}(\Sigma_i^{\text{pos}}) \geq c_{\text{pos}}$, then whenever*

$$\gamma\,\kappa_i^{\min} \ \geq \ \frac{1}{N\tau^2}\Big(\lambda_{\max}(\Sigma_i^{\text{neg}}) - c_{\text{pos}}\Big),$$

Table 4: The statistics of experimental datasets

| Dataset | Clients | Sample | Modality | Class | Dimension |
|---|---|---|---|---|---|
| MNIST-USPS | 24 | 5000 | 2 | 10 | [784,256] |
| BDGP | 12 | 2500 | 2 | 5 | [1750,79] |
| Multi-Fashion | 48 | 10000 | 3 | 10 | [784, 784, 784] |
| NUSWIDE | 24 | 5000 | 5 | 5 | [65, 226, 145, 74, 129] |

Table 5: Number of parameters and runtime by AFMCC.

| Dataset | MNIST-USPS | BDGP | Multi-Fashion | NUSWIDE |
|---|---|---|---|---|
| GPU Memory Usage | 1090MiB | 896MiB | 2072MiB | 902MiB |
| Runtime | 7.6min | 52.9s | 2.0h | 3.2h |

*the dynamics exhibit contraction along the tangent directions at anchor $\boldsymbol{h}_i$, thereby preventing shear splitting.*

Here $\Sigma_i^{\mathrm{neg}} = \sum_j \pi_{ij}^v \boldsymbol{z}_j^v \boldsymbol{z}_j^{v\top}$, $\Sigma_i^{\mathrm{pos}}$ denotes the contribution of positive samples, and $\kappa_i^{\min}$ is the minimum eigenvalue of $\mathcal{K}_i$. It is important to note that this condition relies on the approximation that the softmax distribution is close to one-hot, and it neglects higher-order terms arising from derivatives of the projection operator.

## A.8 DISCUSSION

The above condition demonstrates that the "stiffness" provided by class constraints, quantified by $\gamma \kappa_i^{\min}$, is sufficient to counteract the largest eigenvalue of the negative-sample tensor $\Sigma_i^{\mathrm{neg}}$. Furthermore, the explicit appearance of $1/\tau^2$ indicates that lower temperatures require proportionally larger values of $\gamma$ to maintain stability.

# B EXPERIMENTAL SETTINGS

## B.1 DATASETS

We evaluate our method on four representative multi-modal benchmarks. MNIST-USPS (Peng et al., 2019) is a classic handwritten digit dataset (0–9) with two image modalities: 5,000 paired samples, where MNIST images are $28 \times 28$ and USPS images are $16 \times 16$. BDGP (Cai et al., 2012) contains 2,500 Drosophila embryo images, each represented by a 1750-dimensional visual feature vector and a 79-dimensional textual description. Multi-Fashion (Xiao et al., 2017), following the three-modal construction in (Cui et al., 2023), consists of 30,000 fashion product images, where each sample corresponds to three style variations of the same category, yielding a three-modal dataset with images of size $28 \times 28$. NUS-WIDE (Chua et al., 2009) is a large-scale web image dataset providing multiple feature modalities, including a 65-D color histogram, 226-D block-wise color moments, 145-D color correlogram, 74-D edge orientation histogram, and 129-D wavelet texture. All high-dimensional features are flattened into one-dimensional vectors during preprocessing. For consistency, we randomly sample 5,000 instances across datasets for experimental analysis.

## B.2 SUPPLEMENTARY EXPERIMENTAL DETAILS

All models are implemented in PyTorch (Paszke et al., 2019) and trained on an NVIDIA RTX-4090 GPU. We use ReLU activations (Glorot et al., 2011) and the Adam optimizer. For all datasets, the learning rate is fixed to 0.0003 with a batch size of 1024.

### B.3 COMPARISON METHODS

We compare our approach against nine state-of-the-art methods: HCP-IMSC (Li et al., 2022), IMVC-CBG (Wang et al., 2022), DSIMVC (Tang & Liu, 2022), LSIMVC (Liu et al., 2022), ProImp (Li et al., 2023), JPLTD (Lv et al., 2023), CPSPAN (Jiang et al., 2024), FedDMVC (Chen et al., 2023), FCUIF (Ren et al., 2024), and FMCSC (Chen et al., 2024). Among them, FedDMVC, FCUIF, and FMCSC are federated multi-modal clustering (FedMVC) approaches, while the others are centralized incomplete multi-modal clustering methods.

For a fair comparison, we simplify the heterogeneous mixed-modal setting into a mixed-modal scenario. Specifically, data distributed across clients are concatenated and treated as centralized input (Figure 5). In this setup, samples from multi-modal clients are considered complete, whereas those from single-modal clients are treated as incomplete. Importantly, the reported results reflect that our method is evaluated under the heterogeneous mixed-modal scenario, while competing methods are evaluated under the simplified mixed-modal setting.

Although existing approaches can sidestep heterogeneity by directly concatenating raw data, such practices risk exposing sensitive information and may discourage data owners from participating. In contrast, our method learns complementary clustering structures across clients without revealing raw data, thereby offering stronger privacy guarantees alongside superior performance.

## C  ADDITIONAL EXPERIMENT RESULTS

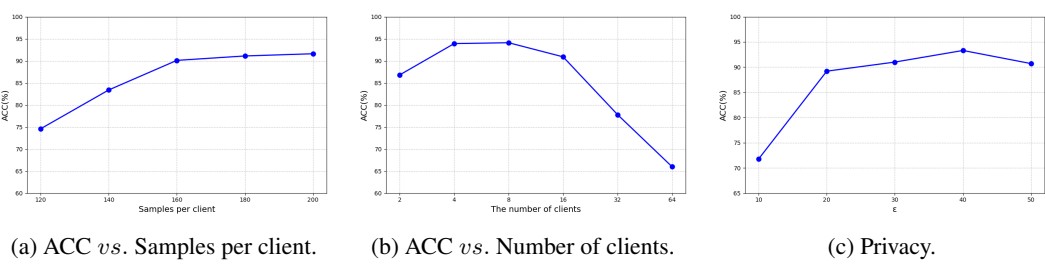

(a) ACC $vs.$ Samples per client.   (b) ACC $vs.$ Number of clients.   (c) Privacy.

Figure 5: Attributes of Federated Learning.

**Samples per client.** We investigate the impact of varying the number of samples per client, as shown in Figure 5a. Increasing the number of samples per client effectively enlarges the overall training data, thereby enhancing the generalization capability of the model.

**Number of clients.** We further study the effect of varying the total number of clients, as illustrated in Figure 5b. A clear performance degradation emerges when the number of clients exceeds 32, which can be attributed to the insufficient number of samples per client.

**Privacy.** By design, AFMCC does not share any raw data between clients and the server. Only the model parameters on each client are transmitted to the server. To further safeguard client privacy, we adopt differential privacy (Abadi et al., 2016) by injecting noise into the parameters uploaded by clients. Figure 5c illustrates the clustering accuracy of AFMCC under different privacy bounds $\varepsilon$. We observe that AFMCC achieves both high performance and privacy when $\varepsilon = 40$.

## D  LARGE LANGUAGE MODELS

In the preparation of this manuscript, large language models (LLMs) were utilized to assist in language refinement and grammatical consistency. Specifically, the models were employed for tasks such as improving sentence fluency, ensuring adherence to academic style, and detecting possible syntactic errors. It is important to note that the LLMs were not used to generate original research content, ideas, or results. All scientific contributions, analyses, and interpretations presented in this work are solely those of the authors.

The use of LLMs in this context is comparable to employing advanced proofreading tools: they provide suggestions and corrections that enhance readability and clarity, while the final decisions

on wording and phrasing remain with the authors. This approach helps maintain the scientific integrity of the work, while ensuring that the manuscript communicates effectively to an international audience.

## E    REPRODUCIBILITY STATEMENT

We are committed to ensuring the reproducibility of our work. To this end, we will release the complete source code and scripts used to train and evaluate our models as supplementary material, along with datasets. The code will be made publicly available in an anonymized repository during the review process and in a permanent repository upon publication. All datasets used in our experiments, including any pre-processing steps, are described in detail in the Appendix and supplementary material, ensuring that others can replicate our results. Hyperparameter settings, model configurations, and evaluation protocols are documented in both the main paper and the supplementary material. Collectively, these efforts ensure that the claims and results presented in this paper can be independently verified and reproduced.

## F    BROADER IMPACTS

Heterogeneous and hybrid modalities are common in real-world applications. Our method extends FedMVC to asynchronous settings, enabling use in domains such as healthcare and IoT where modalities and computational resources are often unevenly distributed. For example, hospitals in developed regions may combine CT, X-ray, and EHR data, while rural clinics rely on a single source. Similarly, smartphones can capture both audio and images, whereas simple recording devices provide only audio. Large organizations may train efficiently, while smaller ones face higher latency. Beyond the well-recognized risks of federated learning, we do not anticipate additional negative societal impacts from this work.

## G    LIMITATIONS

Our model shows strong performance under data heterogeneity, missing modalities, and unbalanced client capacity. However, it assumes balanced class distributions, which may not hold in practice where categories are often imbalanced. Future work will relax this assumption and adapt the framework to more realistic settings.

