# OpenReview forum: "AFMCC: Asynchronous Federated Multi-modal Constrained Clustering"
_ICLR.cc/2026/Conference — Submitted to ICLR 2026_

### Official Review · Reviewer_ra9y · 2025-10-30

**Soundness:** 2
**Presentation:** 2
**Contribution:** 2
**Rating:** 2
**Confidence:** 3

**Summary:**

This paper addresses the issues of representation degradation, modality absence, and imbalance in federated multimodal clustering by proposing an asynchronous federated multimodal constrained clustering method, referred to as AFMCC. This method prevents representation degradation and enhances the separability of multimodal data clustering by calculating the Class-Correlation Matrix $Q$ between different categories and integrating it with the loss function of the target matrix $Q_{tgt}$. Additionally, the article designs a client-specific weighted aggregation approach to effectively handle the problem of modality absence. Experimental results from various benchmark tests demonstrate that AFMCC outperforms other methods in terms of performance.

**Strengths:**

1. This paper addresses the issues of modality absence and representation degradation in multimodal clustering, which holds significant research value.
2. ﻿It also provides open-source code and datasets, offering strong support for community development.

**Weaknesses:**

The innovation of this method is relatively limited. While the design of the Class-Correlation Matrix (CCM) is interesting, the calculation details of the target matrix are unclear. What is the specific process for calculating $Q_{tgt}$? What is the difference between $P_{ai}$ and $P$? Does the calculation of $Q$ require that all categories of data be present in each client or batch? Additionally, the design of the weighted aggregation has a high computational complexity, as it requires all other clients' models to compute features locally. Assigning higher weights to clients with poor cross-modal feature alignment appears unreasonable and requires further explanation. Furthermore, the design addressing asynchronous improvements appears lacking, making it difficult to effectively resolve issues related to client communication being asynchronous or clients going offline.

**Questions:**

1. The details of the formulas in the article need further modification to improve readability. For example, the matrix $A$ in line 207, the calculation logic of $Q$ in line 248, and the definitions of $Q_{ab}$ and $I_K$ in line 253 are not sufficiently clear. These formulas require further explanation.
2. This paper requires further clarification on how it addresses asynchronous aggregation and alleviates computational imbalance issues.

---

### Official Review · Reviewer_WJyL · 2025-10-30

**Soundness:** 1
**Presentation:** 1
**Contribution:** 2
**Rating:** 2
**Confidence:** 4

**Summary:**

This paper explores a new insight about the reasons why contrastive learning fails in federated clustering, especially when different clients observe different modality subsets. To address this challenge, the authors introduce a new constraint mechanism to avoid clustering degeneration over time, as well as a new client-specific aggregation method.

**Strengths:**

* Explaining contrastive learning in both probabilistic and particle view is interesting
* Extensive baselines and benchmark datasets.

**Weaknesses:**

* Confusing writing. Introduction and related work fail to clarify the motivation and the problem. It is unclear between federated multimodal learning and federated clustering from the authors’ writing.
* Lack of motivation. It is unclear why we need to solve this problem. What is the difference between multimodal clustering in centralized and federated settings? What are the benefits of federated clustering?
* Unclear problem formulation. What is $K$ in line 160, how to determine this number. Why do we enforce these clusters to be balanced, while in the standard settings, clusters can be various in size based on the data distribution.
* Lack of literature review. Contrastive learning - based regularization is used commonly in multimodal learning with different variants[1,2], which are theoretically guaranteed. Why these regularizations can not handle the clustering tasks, since these regularizations are designed for clustering modalities implicitly to perform downstream tasks. The authors should expand their literature reviews to highlight their contributions.
* Lack of contributions. The proposed method, while adding explanation and new insights about federated clustering, seems to be an improvement of FMCSC[3]. Empirically, Figure 3b shows that the constraint loss – one of main contributions – does not affect the performance significantly.

[1] Nguyen et al., Learning Reconfigurable Representations for Multimodal Federated Learning with Missing Data, NeurIPS’25

[2]  Nguyen et al., Fedmac: Tackling partial-modality missing in federated learning with cross modal aggregation and contrastive regularization. NCA’24

[3] Chen et al., Bridging Gaps: Federated Multi-View Clustering in Heterogeneous Hybrid Views, NeurIPS’24

**Questions:**

See Weaknesses

---

### Official Review · Reviewer_vPCD · 2025-10-31

**Soundness:** 2
**Presentation:** 3
**Contribution:** 2
**Rating:** 4
**Confidence:** 4

**Summary:**

The paper proposes AFMCC for federated multimodal unsupervised clustering with (i) a Class-Correlation Matrix (CCM) constraint that projects features into a pseudo-probability space and penalizes deviations from a relaxed target;(ii)a client-specific weighted aggregation, and (iii) asynchronous training to tolerate heterogeneous compute. Experiments on several benchmarks report ACC/NMI/ARI gains.

**Strengths:**

1.Broad problem surface: the method tries to address degeneration in contrastive clustering, missing modalities, and client asynchrony in one framework.
2.The experimental results are of excellent performance, leading in multiple indicators across various datasets.

**Weaknesses:**

1.Limited novelty; heavy rebranding. The “particle-dynamics three-force” story largely repackages standard contrastive attraction/repulsion plus a global regularizer, and relies on strong approximations.
2.Unrealistic core assumptions. The CCM derivation assumes equal class sizes and a known K (see the definition of Q and text around Eq. 4), with no robustness analysis under heavy class imbalance or unknown K—both common in FL.
3.There is no reporting of wall-clock time, communication rounds, bandwidth, or staleness-vs-accuracy curves—so the claimed training-time reduction remains unsubstantiated.

**Questions:**

1.What are the mathematical properties of aggregated weights? How are the weights of Equation (6) constructed from the deviation quantities?
2. How does the CCM behave under long-tail and cross-client imbalance? Can you couple AFMCC with non-parametric clustering when K is unknown?
3.Why is this algorithm effective and can it provide a more convincing theoretical proof.

**Details Of Ethics Concerns:**

As mentioned above.

---

### Official Review · Reviewer_G3Dk · 2025-11-01

**Soundness:** 3
**Presentation:** 3
**Contribution:** 2
**Rating:** 4
**Confidence:** 5

**Summary:**

The paper presents an asynchronous federated multi-modal constrained clustering, which adapts to scenarios with arbitrary missing modalities. This method directly fuses multimodal embeddings into a shared embedding by weighted aggregation. By introducing a class-correlation matrix, it alleviates the degradation of contrastive learning in multimodal clustering. Extensive experiments are performed and detailed theoretical analyses are provied.

**Strengths:**

1. The paper is well-written, and the motivations of the work are clear;
2. Theoretical analyses are solid;
3. The overall design is reasonable.

**Weaknesses:**

1. The introduction of the class-correlation matrix seems to be adopted by several works, which limits the novelty.;
2. The main solution to the arbitrary modality missing problem is to aggregate view-specific embeddings with a calculated weight, which seems somewhat trivial.
3. Experiments seem insufficient.
4. Some texts in the figures are small.

**Questions:**

1. Do the experiments consider the Non-iid distribution of data?
2. Does the method (Section 3.2 in Page 5) require the equal distribution of samples in each cluster? In practice, the data distribution of each client might be highly different (i.e., the Non-IID issue). As a result, even with a loose constraint based on the class-correlation matrix, whether the equal distribution is reasonable in practice should be discussed.
3. The problem statament sets the sample number of each client is $N$, which might not be proper and should be corrected.

---

### Meta-Review · Area_Chair_UvCx · 2026-01-07

**Summary:**

This paper proposes an asynchronous federated multi-modal clustering method by fusing view-specific embeddings with client-specific weights and a class-correlation constraint. It brings the practical issues of modality absence, representation degradation and asynchronous updates into one coherent pipeline. However, the novelty is limited and experiments should be enhanced, which are also agreed by the reviewers. I tend to reject this paper.

**Reviewer Concerns:**

No rebuttal. The novelty and benchmark datasets used in the experiments are outstanding.

**Reviewer Scores:**

All reviewers tend to reject. No rebuttal.

---

### Decision · Program_Chairs · 2026-01-26

Reject